# Molecular Dynamics Simulations of HPr Proteins from a Thermophilic and a Mesophilic Organism: A Comparative Thermal Study

**DOI:** 10.3390/ijms24119557

**Published:** 2023-05-31

**Authors:** Ana K. Gómez-Flores, Edgar López-Pérez, Salomón J. Alas-Guardado

**Affiliations:** 1Departamento de Ciencias Naturales, Universidad Autónoma Metropolitana Unidad Cuajimalpa, Mexico City 05300, Mexico; nerakgomez890@gmail.com; 2Posgrado en Ciencias Naturales e Ingeniería, Universidad Autónoma Metropolitana Unidad Cuajimalpa, Mexico City 05300, Mexico; edgarlopezperez07@gmail.com

**Keywords:** *Bst*HPr protein, thermal stability, molecular dynamics, non-covalent interactions, salt bridges

## Abstract

The histidine-containing phosphocarrier (HPr) is a monomeric protein conserved in Gram-positive bacteria, which may be of mesophilic or thermophilic nature. In particular, the HPr protein from the thermophilic organism *B. stearothermophilus* is a good model system for thermostability studies, since experimental data, such as crystal structure and thermal stability curves, are available. However, its unfolding mechanism at higher temperatures is yet unclear at a molecular level. Therefore, in this work, we researched the thermal stability of this protein using molecular dynamics simulations, subjecting it to five different temperatures during a time span of 1 μs. The analyses of the structural parameters and molecular interactions were compared with those of the mesophilic homologue HPr protein from *B. subtilis*. Each simulation was run in triplicate using identical conditions for both proteins. The results showed that the two proteins lose stability as the temperature increases, but the mesophilic structure is more affected. We found that the salt bridge network formed by the triad of Glu3-Lys62-Glu36 residues and the salt bridge made up of Asp79-Lys83 ion pair are key factors to keep stable the thermophilic protein, maintaining the hydrophobic core protected and the structure packed. In addition, these molecular interactions neutralize the negative surface charge, acting as “natural molecular staples”.

## 1. Introduction

Temperature is one important variable in the natural environment. The classification of the organisms based on temperature is distributed into four groups, psychrophiles, mesophiles, thermophiles, and hyperthermophiles [1,2,3]. Although organisms live under different temperature conditions, they are made up of similar molecular components that allow them to perform vital functions, such as respiration, motion, reproduction, and nutrition [4].

Biomolecules, such as proteins, are part of these essential components and, according to their biochemical functions, they can be classified, in general, as structural, contractile, carriers, and enzymes. The functional diversity of proteins originated from the chemical variety of their building blocks (amino acids), the flexibility of the polypeptide chain, and the multiplicity of ways in which they reach folding [4,5]. It is well known that the arrangement of amino acids and their chemical nature in the primary structure, the formation of α-helices and β-sheets in the secondary structure, and the construction of a stable tertiary structure, have allowed proteins to be functional in specific conditions of salinity, pressure, pH, and temperature, to name a few [1,2,3].

Moreover, the ordered folding of the α-helices and/or β-sheets structures improves the intra- and intermolecular interactions, such as hydrogen bonds, hydrophobic contacts, and salt bridges [4]. An increment in these non-covalent-type interactions helps proteins to endure extreme conditions as those mentioned [2]. In particular, the study of resistant proteins to temperature has received a great deal of attention in experimental, theoretical, and computational research in recent decades; since this variable is of great interest in the improvement of thermostable enzymes with potential industrial and biotechnological applications [2,6,7,8,9,10].

In addition to this aspect, there is an intrinsic and fundamental interest to understand at a molecular level the contribution of each non-covalent interaction to the thermal stability of such biomolecules. Different computational tools have been used for this purpose, in particular, molecular dynamics (MD) simulation methods have provided relevant information about these interactions, as has been reported in the literature [2,9,10,11,12,13,14,15,16]. An interesting strategy consists of directly comparing structural and physicochemical behaviors between homologous proteins, for example, a thermophilic protein and its mesophilic counterpart. Thus, MD approaches have commonly shown that the thermophilic proteins increase their non-covalent interactions, reduce the number of charged residues on the surface, and maintain the structure together and packed [11,12,13,14,15,16].

Homologous proteins can have a high percentage of identical and similar amino acids in their sequences. For example, the histidine-containing phosphocarrier protein (HPr) from the thermophilic organism *Bacillus stearothermophilus* (*Bst*HPr) has 70% sequence identity and 85% sequence similarity with the HPr protein from the mesophilic organism *Bacillus subtilis* (*Bs*HPr), exhibiting very similar tertiary structures [17,18,19]. However, the melting temperature (Tm) differs between the two proteins by about 15 °C, i.e., the *Bst*HPr protein presents higher temperature resistance and 50% of its structural unfolding occurs at 88.9 °C, whereas the Tm of the *Bs*HPr protein occurs at 74.4 °C [17,18]. Thus, the amino acids’ chemical nature and spatial arrangements can cause one protein to decrease or increase its stability due to temperature changes.

For these reasons, and because of the biomolecular importance of the HPr protein in the catabolism of different bacterial organisms, in this work the thermal stability of the *Bst*HPr protein is analyzed and compared with its homologue *Bs*HPr using molecular dynamics simulations. For this purpose, different structural parameters and molecular interactions were studied in both variants. It is worth mentioning that the HPr protein is an essential component of the bacterial phosphoenolpyruvate:sugar phosphotransferase system (PTS) that acts as a global regulator of signal transduction. This complex controls physiological processes related to prokaryote nutrition during bacterial growth [20,21].

## 2. Results and Discussion

The results are divided into three subsections. First, we explore the structural behaviors such as secondary structure profiles, RMSD, Rg, and native contacts. Next, we analyze molecular interactions such as hydrogen bonds, hydrophobic contacts, and salt bridges. Finally, we investigate the electrostatic surface potential. The details are given as follows.

### 2.1. Structural Behaviors

#### 2.1.1. Secondary Structure Profiles

Figure 1 shows the secondary structure profile of the *Bst*HPr protein at 298 K and 1 μs simulation time. This profile is organized according to the protein’s primary sequence, starting with the N-terminal domain, which contains the secondary structures β_1_, α_1_, β_2_, and β_3_, and ending with the C-terminal domain, which includes the structures α_2_, β_4_, and α_3_.

Figure 2 shows the secondary structure profiles of both proteins for simulation 1 at five different temperatures (Appendix A show such profiles for the two replicas). In this way, the two proteins maintain conformational stability at 298 K. However, the α_2_-helix structure in the protein *Bst*HPr is lost after 500 ns of simulation when the temperature increases up to 333 K. This structure is lost for both proteins at 362 K, but in the *Bs*HPr protein, this occurs from the start of the simulation. In general, the α_2_-helix structure is very unstable and sensitive to temperature changes in both proteins, but the thermophilic protein presents greater thermal instability. To corroborate this effect, we have performed Ramachandran plots of this secondary structure at 298 and 333 K. Additionally, we carried out such analysis for the α_3_-helix structure at 298 K, which remains stable for both proteins, but the thermophilic protein displays greater stability than the mesophilic one. These details can be found in the Appendix A.

Both proteins lose structural content along the simulations as the temperature increases. For example, at 362 K, the *Bst*HPr protein loses more α_1_-helix structure compared with its *Bs*HPr counterpart; moreover, this transforms to a π-helix structure at the end of the simulation. Notice that, although the temperature modifies the spatial arrangement of the proteins, the anti-parallel structure of the β-strands is conserved at this temperature. Moreover, the α_1_-helix is lost at ≈200 ns, the α_2_-helix is not formed, and the α_3_-helix is lost at ≈500 ns for the two proteins at 400 K. A combination of α-helix structures is formed during the trajectories, i.e., remnants of α-helix structures remain and 3_10_-helix arrangements are created. These α-helices can even be transformed into β structures, which occurs mainly in the *Bs*HPr protein, e.g., β strands are generated at ≈450 ns in the α_1_ and α_2_ arrangements. At this temperature, the β structures are more affected in the mesophilic protein compared to its thermophilic homologue pair, since these are lost at certain time ranges. For instance, the β_1_ and β_4_ strands are lost between ≈800 ns and ≈940 ns, while the β_2_ and β_3_ strands are lost after ≈850 ns. In general, disordered structures, such as random coil, bend, and turn are formed when both proteins lose α-helices and β strands.

Finally, the *Bst*HPr protein maintains fragments of secondary structures more stable than its mesophilic counterpart at 450 K. For example, the adjoining β_2_ and β_3_ strands remain stable during almost the entire trajectory and the α_3_-helix is rarely present at ≈850 ns of simulation; afterward, they disappear. During this process, the β_1_ strand is the only stable structure in the *Bs*HPr protein. It is important to consider that at this stress temperature, both proteins would lose their native secondary structures completely. However, the thermophilic protein maintains almost intact the β_2_ and β_3_ strands throughout the simulation.

Additionally, Figure 2 shows one structural conformation for each secondary structure profile for both proteins, which was taken at 750 ns of simulation. These conformations exhibit the unfolding processes of the two protein structures by temperature effects. It is possible to observe that the β_1_, β_2_, and β_3_ strands of *Bst*HPr protein and α_3_-helix of *Bs*HPr protein are conserved at this time step at 450 K.

Similar behavior can be observed in the secondary structure profiles of simulations 2 and 3. In general, the β strands are more stable in *Bst*HPr protein in comparison with its mesophilic homologue one, when the temperature is elevated at 450 K. These facts can be consulted in the Appendix A.

#### 2.1.2. Root Mean Square Deviation

In Table 1, the average RMSD values (avg) and their standard deviation values (SD) are shown for both *Bst*HPr and *Bs*HPr proteins at five temperatures of interest, while Figure 3 shows the RMSD trajectories for three simulation temperatures: 298, 362, and 400 K.

Table 1 indicates that the average RMSD values for *Bst*HPr and *Bs*HPr proteins are 0.102 ± 0.040 nm and 0.121 ± 0.024 nm at 298 K, respectively. Figure 3a shows the behavior of these trajectories. These results indicate that both proteins’ structures remain stable during the simulation time, but the thermophilic protein presents more atomic fluctuations than the mesophilic protein, since the SD value of the *Bst*HPr protein is slightly greater than that of the *Bs*HPr protein. This is in contrast to the usual hypothesis that conformational rigidity determines thermal stability. However, Karshikoff et al. [22] have shown that with increasing available configurations, the thermophilic protein tm_gdh becomes more stable, relative to its mesophilic counterpart.

However, these fluctuations are reverted as the temperature increases, i.e., the average and standard deviation RMSD values for the *Bst*HPr protein are lower than those found in *Bs*HPr protein at 333, 362, and 400 K, which can be found in Table 1 and Figure 3b,c. For example, these values are 0.173 ± 0.058 nm and 0.214 ± 0.095 nm for *Bst*HPr and *Bs*HPr structures at 362 K, respectively. This behavior indicates that the thermophilic protein’s structure is more stable than that of its mesophilic homologue pair, suggesting that temperature impacts cause the *Bs*HPr structure to go through additional conformational states.

When both proteins reached the temperature of 450 K, their RMSD values increased by almost one order of magnitude with respect to their native state (298 K), yielding average values of 1.133 ± 0.437 and 1.225 ± 0.322 nm for *Bst*HPr and *Bs*HPr protein, respectively. Although the RMSD average value of *Bst*HPr protein is the lowest, it presents more fluctuations with respect to the *Bs*HPr protein, because its standard deviation is greater (0.437 nm). Nevertheless, both structures lose stability, and thus unfold at this temperature.

#### 2.1.3. Radius of Gyration

The average Rg values and their standard deviation values for the *Bst*HPr and *Bs*HPr proteins at the five temperatures of study are shown in Table 2 and the evolution of this parameter for three simulated temperatures 298, 362, and 400 K is shown in Figure 4.

The two proteins practically present the same average Rg value at the temperature of 298 K, that is, 1.181 ± 0.007 nm for *Bst*HPr and 1.185 ± 0.007 nm for *Bs*HPr, and they maintain their native state during the entire simulation time (Figure 4a). This value slightly increases for both proteins at 333 K, reaching the same average value, which changes on the third significant digit after the decimal point as in 298 K: 1.195 ± 0.013 nm for *Bst*HPr and 1.193 ± 0.013 nm for *Bs*HPr.

At the temperature of 362 K, the proteins do not undergo changes in the average Rg values compared with 333 K, in these cases, the values are 1.190 ± 0.010 nm for *Bst*HPr and 1.195 ± 0.014 for *Bs*HPr. A slight structural expansion is observed around 700–800 ns in the mesophilic variant (Figure 4b).

When increasing the temperature up to 400 K, the mesophilic protein has an average Rg value of 1.241 ± 0.046 nm and shows greater flexibility, e.g., there is a structural expansion around 200–500 ns, after 600 ns it is compacted returning to the Rg value of the native state; it even undergoes compactification beyond this state, and at the end of the simulation the protein structure is expanded once again. Meanwhile, the thermophilic protein has an average Rg value of 1.213 ± 0.023 nm and presents slight structural fluctuations, but these are lower than those observed in the *Bs*HPr protein (Figure 4c).

Finally, if temperature is raised up to 450 K, both proteins increase the average Rg values with respect to their native states. These values are 1.389 ± 0.228 nm for *Bst*HPr and 1.415 ± 0.188 nm *Bs*HPr. Consequently, the proteins undergo constant structural expansions and contractions during all the trajectories.

#### 2.1.4. Native Contacts

Table 3 shows the fraction of the average native contacts values and their standard deviation values for the two proteins at the five temperatures of interest, while Figure 5 depicts the trajectories at 298, 362, and 400 K. The initial value of native contacts is equal to 1 at any temperature, that is, the native contacts are formed between topological atoms of the first conformation in t = 0 ns, and this fraction decreases during the simulation times.

At the temperature of 298 K the average value of the fraction of native contacts for both proteins is almost the same: the value is 0.938 ± 0.029 for the thermophilic variant and 0.925 ± 0.029 for its mesophilic counterpart. The trajectory corresponding to the *Bst*HPr protein presents a slight soft fall, but the behavior of the curve of the *Bs*HPr protein shows some fluctuations, which indicates that native contacts break and then are reorganized; moreover, these contacts fall around 900 ns of simulation (Figure 5a).

At 333 K, the average values of the fraction of native contacts for the *Bst*HPr and *Bs*HPr proteins are 0.887 ± 0.047 and 0.870 ± 0.073, respectively. Both proteins lose around 6% on average of native contacts, that is, with respect to their contacts at 298 K. The mesophilic protein has two significant falls at ≈300 ns and ≈900 ns of simulation, while the thermophilic one presents the most notable fall around 400–500 ns (see Appendix A).

When the system is exposed to a temperature of 362 K, the *Bst*HPr protein keeps its native contacts constant for almost the whole trajectory and its average value is essentially equal to the temperature of 333 K (0.884 ± 0.054). At the same time, the average value of native contacts in *Bs*HPr protein is 0.832 ± 0.101, indicating that these contacts fall on average ≈10% in comparison with the native structure. The most important falls occur around 200 ns; after that the structure stabilizes, turning to a considerably decrease again at ≈700 ns (Figure 5b).

When increasing the temperature to 400 K, the average values of the fraction of native contacts for both *Bst*HPr and *Bs*HPr variants are 0.662 ± 0.164 and 0.513 ± 0.162, respectively. The native contacts of the mesophilic protein decrease on average by about 45% and the trajectory presents two salient falls, the first occurs from 0 to ≈200 ns and the second around 800 ns of simulation. On the other hand, the thermophilic protein loses around 29% of native contacts and the most significant fall occurs from ≈580 ns. It is noteworthy that the curves of both proteins have a steeper slope with respect to those observed in their native structures, i.e., at 298 K (Figure 5c).

At the temperature of 450 K, the thermophilic protein has an average value of the fraction of native contacts of 0.199 ± 0.173 and its counterpart presents an average value of 0.142 ± 0.131. At this temperature, the proteins lose approximately 78% (*Bst*HPr) and 85% (*Bs*HPr) on average of native contacts. As can be observed in Appendix A, the *Bst*HPr protein retains a higher fraction of these contacts.

### 2.2. Molecular Interactions

#### 2.2.1. Hydrogen Bonds

Table 4 shows a summary of the average and standard deviation values of hydrogen bonds (HB) corresponding to the residues buried in the protein, known also as protein–protein interactions (labeled as HBpp), and the residues of the protein surface with the water, known as protein-solvent interaction (labeled as HBps). Figure 6 shows the HB simulation trajectories at 298, 362, and 400 K.

Table 4 and Figure 6 show that both types of interactions, HBpp and HBps, decrease as the temperature rises for the two proteins. This is mainly caused by the increase in the kinetic energy of the particles in the system, promoting the breakdown of HBs, since in order for this interaction to be maintained and the hydrogen bond strength to be adequate, particular geometrical conditions must be met [23] (see Appendix A).

Furthermore, Table 4 shows that the thermophilic protein forms slightly more HBpp on average than its mesophilic counterpart at 298 K: 62.4 ± 3.7 for *Bst*HPr and 59.4 ± 3.8 for *Bs*HPr. However, both proteins attain practically the same HBpp value at 450 K: 41.6 ± 7.3 for *Bst*HPr and 41.7 ± 6.7 for *Bs*HPr. Moreover, with the values reported in this table, the amount of HBpp and HBps lost for each temperature from both proteins’ native structures can be calculated.

First, we determined the average value and percentage of HBpp lost from the structures of the *Bst*HPr and *Bs*HPr proteins with respect to 298 K, respectivel;: 2.2 (3.5%) vs. 2.2 (3.7%) at 333 K, 3.9 (6.2%) vs. 2.6 (4.4%) at 362 K, 9.3 (14.9%) vs. 8.7 (14.6%) at 400 K, and 20.8 (33.3%) vs. 17.7 (29.8%) at 450 K. It can be observed that in the temperature interval between 298 and 400 K both proteins lose almost the same percentage of HBpp (14.9% vs. 14.6%), but this percentage increases more for the *Bst*HPr protein than for the *Bs*HPr one in the interval from 298 to 450 K (33.3% vs. 29.8%). If the calculation is performed between 400 and 450 K, it is observed that thermophilic and mesophilic proteins lose 21.7 and 17.8 % of HBpp, respectively. The thermophilic protein loses more HBpp up to 450 K because it forms slightly more HBpp interactions compared to its mesophilic homologue pair at 298 K.

Secondly, these same measurements were made for the HBps loss, which are for the *Bst*HPr and *Bs*HPr proteins, respectively: 5.0 (2.7%) vs. 6.7 (3.2%) at 333 K, 11.9 (6.4%) vs. 15.1 (7.2%) at 362 K, 14.7 (7.9%) vs. 21.7 (10.4%) at 400 K, and 12.3 (6.6%) vs. 21.3 (10.2%) at 450 K. In this case, the *Bs*HPr protein lightly loses more percentage of HBps than the *Bst*HPr protein between 298 and 400 K. However, at 450 K these interactions slightly enhance, due to the structural arrangements between water molecules and proteins caused by the increased kinetic energy. Moreover, we can observe that in the range between 298 and 450 K the mesophilic variant loses more percentage of HBps compared whit the thermophilic protein (6.6% vs. 10.2%). This occurs because the *Bs*HPr structure forms more HBps with respect to the *Bst*HPr structure at 298 K.

Therefore, both proteins lose HB in the temperature range from 298 to 450 K, but the thermophilic protein forms a greater amount of buried HB; in the meantime, the mesophilic protein forms a greater number of surface HB. These facts contribute to that the thermophilic protein is better packed and undergoes smaller global fluctuations (see Figure 3) and compaction/expansion states (see Figure 4) than those observed in the mesophilic variant at unfolding temperatures.

#### 2.2.2. Solvent Accessible Surface Area

Table 5 shows the corresponding summary of the average and standard deviation values for solvent-accessible surface for area polar (labeled as SASAp) and non-polar (labeled as SASAnp) residues of both proteins. In this way, Figure 7 shows the trends of these parameters at 298, 362, and 450 K.

In its native state, the mesophilic protein presents higher SASAp values with respect to those observed in the thermophilic protein; the average value for the *Bs*HPr protein is 20.43 ± 0.68 nm^2^, while for the *Bst*HPr protein is 18.37 ± 0.72 nm^2^. This is due to the fact that the *Bs*HPr protein has a higher number of polar residues than the *Bst*HPr protein, corresponding to 48.3 and 44.3% of the total residues, respectively.

The SASAp values increase for both proteins when the temperature rises from 298 to 450 K. The thermophilic and mesophilic proteins present average values of 20.89 ± 1.94 nm^2^ and 22.06 ± 1.98 nm^2^ at 450 K, respectively. Therefore, from the native state to the denatured state, the *Bst*HPr protein increases its area by 2.52 nm^2^ and the *Bs*HPr protein increases this area by 1.63 nm^2^, which corresponds to percentage increases of 13.7 and 8.0%, respectively.

Although both proteins expose more of their polar residues to the solvent with increasing temperature, the thermophilic protein exposes a greater polar area than its mesophilic counterpart. This is possible because the mesophilic protein has almost a fully exposed the area of its polar residues to the solvent from 298 K, whereas the thermophilic protein has a greater buried polar area, which is shown to the solvent as the temperature increases.

By contrast, the thermophilic protein presents higher values of SASAnp than those observed in the mesophilic one in the native state; the average value for the *Bst*HPr protein is 30.89 ± 0.74 nm^2^, while for the *Bs*HPr protein it is 28.84 ± 0.80 nm^2^. The average values of SASAnp increase for the two proteins as the temperatures increase too. In the temperature range from 298 to 400 K, the area of the *Bst*HPr protein increases by 2.58 nm^2^ and the area of the *Bs*HPr protein increases by 5.06 nm^2^, corresponding to an increase of 8.4 and 17.6% of each area, respectively. This means that the mesophilic protein undergoes greater exposure of its nonpolar residues to the solvent when it reaches 400 K. This finding is interesting to analyze since the hydrophobic core of the mesophilic protein is less protected, against energetic changes, than that of its thermophilic counterpart.

In addition to this fact, it can be observed that in the temperature range between 298 and 362 K, the areas of the non-polar residues increase by 0.9 and 2.3% for *Bst*HPr and *Bs*HPr proteins, respectively, hence these areas change slightly for both proteins. However, between 362 and 400 K, the areas of the non-polar residues increase considerably by 7.4% for *Bst*HPr and 14.9% for *Bs*HPr. At 450 K, it is observed that the non-polar area increases significantly for the two proteins, whose average values are 42.93 ± 5.94 and 43.22 ± 5.13 nm^2^ for *Bst*HPr and *Bs*HPr, respectively. Therefore, from 400 to 450 K the non-polar area of both proteins increases almost the same percentage, 28.3% for *Bst*HPr and 27.5% for *Bs*HPr, indicating that the hydrophobic core of the proteins has been destabilized.

In the temperature range between the native state and the unfolding state, i.e., from 298 to 450 K, the SASAnp increases by 39.0% for the *Bst*HPr protein and 49.9% for the *Bs*HPr protein. Thus, the thermophilic protein keeps a less unstable hydrophobic core than that of its mesophilic variant. This occurs because of three situations, (1) the *Bst*HPr protein contains a greater number of non-polar residues (aliphatic and aromatic residues), in this case, the structures of the *Bst*HPr and *Bs*HPr proteins have 49 (55.7%) and 45 (51.7%) residues of that nature, respectively; (2) as specified, the *Bst*HPr protein has more buried HB; and/or (3) the *Bst*HPr molecule presents a greater number of salt bridges that protect the hydrophobic core, which will be analyzed in the following Section 2.2.3.

Finally, we performed analyses of the ILV clusters for both proteins at 250, 500, 750, and 1000 ns using the ProteinTools tool [24]. These analyses corroborate that the hydrophobic core of the *Bst*HPr protein is more stable than that of the *Bs*HPr one. Details of these calculations can be found in the Appendix A.

#### 2.2.3. Salt Bridges

Table 6 and Table 7 show the average values of the salt bridge frequencies for the *Bs*HPr and *Bst*HPr proteins using the GetContacts tool at five different temperatures. As stated in the Appendix A, one SB is stable when its frequency is 0.30 during the simulation trajectories. Considering this criterion, we can observe in both tables that *Bs*HPr and *Bst*HPr proteins form 3 and 5 salt bridges in the native state (298 K), respectively.

These tables show that the Glu84–Arg17 salt bridge has a frequency value of around 0.5 at 298, 333, and 362 K temperatures for both proteins. This means that this bridge is conserved and does not break, even if the temperature increases. This fact is probably related to the functional mechanism of the HPr protein, since the Arg17 and His15 residues have a strong relationship with the geometry of the active site. Indeed, the flexibility of the active site is important for the His15 residue to be phosphorylated [25,26]. Then, in addition to stabilizing the HPr structure, the Glu84–Arg17 interaction can affect the active site’s open/closed conformational transitions.

The Asp11–Lys57 salt bridge is also a conserved bridge in both proteins, but it forms more frequently in the *Bs*HPr protein at 298, 333, and 362 K. This salt bridge is favored due to the chemical nature of the Asp11 residue since the interaction between this residue with Lys or Arg is larger and the conformational entropy penalty is lower compared to salt bridges formed by the Glu residue [27]. The formation of this salt bridge decreases with increasing temperature at 400 K in both proteins.

Further to these two conserved salt bridges, each protein has a particular salt bridge. In the mesophilic protein, an inter-structural salt bridge is formed between residues Glu78 and Lys4 belonging to the α_3_-helix and β_1_ strand structures (Figure 8a), respectively, which has an average distance of 1.99 ± 0.39 Å. While in the thermophilic protein, an intra-structural salt bridge formed by residues Asp79 and Lys83 was found in α_3_ helix structure (Figure 8b) with an average distance of 1.89 ± 0.16 Å. It is noteworthy that these calculations were performed from the three independent simulations at 298 K, considering the r_SB_ ≤ 4 Å condition [28]. This distance was used as a criterion for all salt bridge measurements. Kumar and Nussinov asserted that close-range interactions, such as salt bridge pairs, contribute to the stability of helical structures [29]. In this sense, the Asp79-Lys83 salt bridge contributes, due to its sequential closeness, to keep the α_3_-helix structure of the *Bst*HPr protein stable, which was described in Section 2.1.1 (see Figure 2, Appendix A). A fact that corroborates the stability of this salt bridge is that it maintains an average frequency greater than 30% at 450 K.

Different investigations have shown that salt bridge networks are essential to stabilize protein structures [29,30,31,32,33]. As can be seen in Table 7, the thermophilic protein forms a triad of residues made up of the Glu3–Lys62 and Glu36–Lys62 pairs (Figure 8c), which is stable up to 400 K. As these residues are part of the β_1_ (Glu3), β_2_ (Glu36), and β_4_ (Lys62) strands, the triad (Glu3–Lys62–Glu36) causes the β-sheet, formed by the four antiparallel strands, to be more stable in the *Bst*HPr protein with respect to the *Bs*HPr protein when the temperature increases. As explained in Section 2.1.1, remnants of β-strands of the thermophilic protein can be observed in secondary structure analysis at 450 K, which agrees with the profiles shown in Appendix A. As stated by Kumar and Nussinov, the close-range electrostatic interactions, such as salt bridges and their networks in proteins, contribute meaningfully to the difference in protein stability between thermophilic and mesophilic homologues [29]. Therefore, the salt bridge network located in the thermophilic protein represents a fundamental and indispensable interaction that protects the hydrophobic core, causing greater thermal stability and structural maintenance of this biomolecule.

In addition to these facts, the analysis of the charged residues, using the PropKa server, showed that all of them are on the exterior of the two proteins, indicating that the salt bridges have a superficial nature. Consequently, each interaction between ionic pairs acts as a “natural molecular staple” that contributes to maintaining stable the hydrophobic core, corroborating that the triad of Glu3–Lys62–Glu36 residues is essential for the thermophilic protein to tolerate elevated temperatures.

### 2.3. Electrostatic Surface Potential

As all charged residues are placed on the surface of the proteins, the electrostatic surface potential was measured at 298 K. Thermophilic and mesophilic proteins showed a net charge of −1 and −4, respectively, indicating that on the surface of the *Bs*HPr protein there is a higher charge repulsion potential, thereby this structure undergoes greater structural destabilization, since the charge repulsion potentials cause that the proteins lose structural stability as the temperature rises [12].

As stated, both structures have a β-sandwich folding type, which is composed of one β-sheet as opposed to three α-helices. Although the tertiary structures are identical, the ESP maps show meaningful changes between the two proteins. The most important change occurs on the β-sheet surface, since the mesophilic protein has a higher negative ESP, whereas the thermophilic protein has positive zones that neutralize this secondary structure (Figure 9a,d). The surface of the α-helix secondary structures shows the same behavior. In particular, it is observed that the α_3_-helix of the mesophilic protein presents a higher negative ESP than that of the thermophilic protein (Figure 9b,e). In the case of the structural interphase β-sheet/α-helices, there are slight changes, but, in general, the same ESP pattern is observed for both proteins (Figure 9c,f). Both the salt bridge network (Glu3–Lys62–Glu36) and the salt bridge Asp79–Lys83 neutralize the negative charge on the surfaces of the β-sheet and the β_3_-helix of the thermophilic protein, respectively, causing the repulsion potential decreases and thus making this protein more stable at elevated temperatures.

In summary, the negative ESP of the *Bs*HPr protein causes electrostatic repulsive effects that destabilize the structure when the temperature increases [12]. Such repulsive effects can be reduced if there are positively charged residues, as occurs in the *Bst*HPr protein. Directed mutagenesis studies have been performed to increase protein thermostability, which drastically change their ESP by adding a high content of charged residues of the same type [34,35]. However, it is important to note that, in those studies, modified proteins exhibit an additive effect on stability by mutating a large number of residues. On the other hand, the evolutionary (natural) design that occurs between the *Bst*HPr/*Bs*HPr protein pair is more refined, since, with a few substitutions in the charged residues located on the surface, electrostatic interactions such as salt bridges are modulated, in addition to the redistribution of ESP globally.

## 3. Materials and Methods

### 3.1. Molecular Models

The structural data of both HPr variant proteins were taken from the Protein Data Bank (PDB, www.rcsb.org (accessed on 24 April 2023)) of the meso- and thermo-bacteria *Bacillus subtilis* (PDB-id 2HPR) [36] and *Bacillus stearothermophilus* (PDB-id 1Y4Y) [17], respectively. The crystallographic coordinates show that the two structures form an identical open-faced β-sandwich, which is made up of one β-sheet with four antiparallel β-strands and three α-helices following a β_1_α_1_β_2_β_3_α_2_β_4_α_3_ secondary structural configuration (Figure 10) [17,37,38].

### 3.2. Molecular Dynamic Simulations

First, both crystal structures were protonated using the PropKa server [40] at neutral pH = 7.0. Afterward, the MD simulations were performed using the free GROMACS v2020.3 program [41,42] and the AMBER99SB force field [43]. Each protein was placed inside a dodecahedral box, with periodic boundary conditions applied on all faces of the box. The solvent molecules were modeled with the SPCE water model [44] and the distance between the protein and the edge of the box was set equal to 1.5 nm, thus there is a distance of at least 3.0 nm between the protein and its periodic images. Moreover, the *genion* algorithm [41] was used to add suitable counterions to neutralize the total charge of the systems at pH = 7.0. Table 8 displays a list of the conditions of the simulation for both proteins. The two systems (protein, solvent, and ions) are also shown in the Appendix A.

Long-range electrostatic interactions were calculated with the Particle Mesh Ewald (PME) method [45], using a Coulomb cut-off of 1 nm with Fourier spacing of 0.16 nm at 4th order cubic interpolation, while van der Waals interactions were computed using a cut-off of 1 nm. The energy of the systems was minimized using the *steepest descent* algorithm [46]. This process converged in 624 steps, where the maximum force was less than 1000 kJ·mol^−1^·nm^−1^, and the potential energy was −2.68 × 10^5^ kJ·mol^−1^. After that, it was necessary to equilibrate the temperature and pressure in the system. The equilibration of the temperature was completed using an extension of the Berendsen thermostat known as the *velocity-rescaling* method [47]. In this work, the system was equilibrated to all temperatures of study, corresponding to 298, 333, 362, 400, and 450 K. On the other hand, the pressure was equilibrated at 1 bar through the Parrinello–Rahman barostat [48] and one compressibility coefficient of 4.5 × 10^−5^ bar^−1^. Both NVT and NPT ensembles were equilibrated employing 100 ps.

Hereafter, the LINCS algorithm [49] was used to constrain the bond lengths achieving an integration step each 2 fs, thus 500 million steps were simulated during the production stage, resulting in 1 μs of simulation. Particle position and energy data were saved each 100 ps, thus yielding 10,001 system configurations. We carried out three independent simulations for each temperature to assess the reproducibility of the simulations and explore more conformational states of the proteins. Consequently, 30 simulations were run, 15 simulations for each protein. In addition, the mean computer time spent for each simulation was 2 days, 5 h, and 48 min.

The temperatures used in this work were selected according to (a) 298 K corresponds to the temperature nearest to maximal stability (Ts) obtained from stability curves of the two proteins, (b) 348 and 362 K matches with the melting temperatures of *Bs*HPr and *Bst*HPr proteins, respectively [18], and (c) 400 and 450 K are “stress” temperatures as unfolding pathways for both proteins, i.e., these temperatures have been proposed to activate the protein unfolding since the timescale of this process between experiments and simulations are very different, as stated by Daggett et al. [50].

### 3.3. Simulation Analysis

In this work, we determined the following structural parameters in both proteins, (a) secondary structure profiles obtained with the algorithm *define secondary structures of proteins* (DSSP) [51], (b) global fluctuations through the root mean square deviation (RMSD) [52,53], (c) the global packaging using the radius of gyrations (Rg) [54], and (d) the fraction of native contacts (Q) by mean of MDTraj program [55,56]. Moreover, we calculated the molecular interactions, (a) hydrogen bonds (HB) using the *hbond* algorithm of GROMACS [41], (b) salt bridges (SB) employing the GetContacs tool (https://getcontacts.github.io/ (accessed on 16 March 2023)) and *mindist* function of GROMACS [41], and (c) hydrophobic contacts through the solvent accessible surface area (SASA) with the *freeSASA* algorithm [57,58]. Finally, the electrostatic surface potential (ESP) was determined using the Adaptive Poisson–Boltzmann Solver (APBS) package [59,60]. Details about these analyses can be found in the Appendix A.

## 4. Conclusions

In this work, we analyzed the thermal stability of the thermophilic protein *Bst*HPr using molecular dynamics simulations at five different temperatures for 1 μs time period. To achieve this, we measured structural parameters and molecular interactions that were compared with those of the *Bs*HPr mesophilic protein. The comprehensive analysis of these measurements sheds light on the stability of the *Bst*HPr protein under the effect of temperature.

We found that the native state structures (298 K) of the two proteins do not show significant differences. As the temperature increases, both proteins undergo changes in their secondary structures, but the mesophilic protein is the most affected. The thermophilic protein conserves the β-sheet and the α_3_-helix more unaffected, even at 450 K. Furthermore, it maintains the β_2_ and β_3_ strands stable during almost the entire simulation trajectory.

The RMSD, Rg, and Q values allowed us to ascertain the structural differences between the *Bst*HPr and *Bs*HPr proteins. The global fluctuations and the compact/expansion states are lower in the thermophilic protein, while the mesophilic protein loses more native contacts.

Analysis of molecular interactions shows that the thermophilic protein, compared with its mesophilic counterpart, (1) forms more buried HB and less surface HB, but with increasing temperature, it loses a higher percentage of HBpp and a lower percentage of HBps; (2) has a lower polar area that is more exposed to solvent but exhibits higher nonpolar area that is less exposed to solvent if the temperature increases, indicating that its hydrophobic core is more stable; and (3) contains a greater number of salt bridges that help it resist temperature changes. In particular, the structural arrangements formed by the triad of Glu3–Lys62–Glu36 residues and the Asp79–Lys83 ion pair, that keep the β-sheet and α_3_ helix stables, function as “natural molecular staples” protecting the hydrophobic core and contributing to structural packing.

Additionally, we calculated that 100% of charged residues are located on the surfaces of both proteins at 298 K. These residues produce net charges of −1 and −4 for the *Bst*HPr and *Bs*HPr proteins, respectively, causing higher charge repulsion potential in the structure of the latter protein. The thermophilic protein reduces its negative charge by the presence of the salt bridge network (Glu3–Lys62–Glu36) and the salt bridge (Asp79–Lys83), which are key interactions to maintain its thermal stability at elevated temperatures.

Finally, in order to corroborate the important role played by the Glu3–Lys62–Glu36 triad in the thermal stability of *Bst*HPr protein, we have performed the mutation of the Lys62 residue by Ala62. However, such results will be reported and discussed in future work.

## Figures and Tables

**Figure 1 ijms-24-09557-f001:**
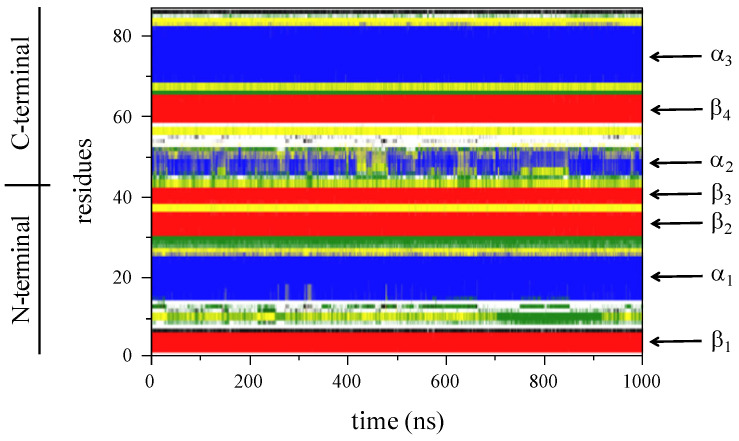
Secondary structure profile of the *Bst*HPr protein at 298 K obtained from the DSSP algorithm. The color code of the structures is: random coil (white), β-sheet (red), β-bridge (black), bend (green), turn (yellow), α-helix (blue), π-helix (purple), and 3_10_-helix (gray). The labels are identical to those in panel c of the figure in Section 3.1.

**Figure 2 ijms-24-09557-f002:**
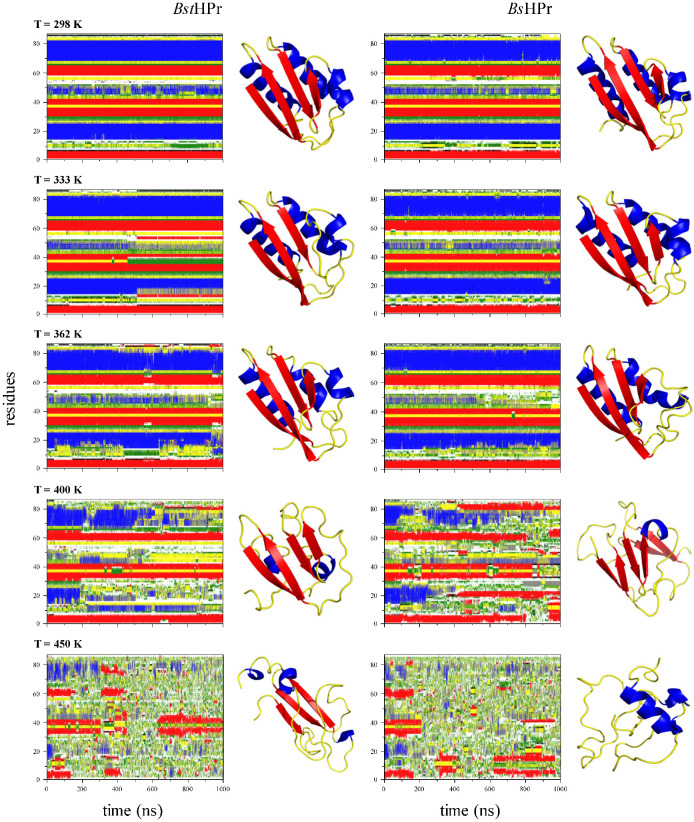
Secondary structure calculations based on the DSSP algorithm for *Bst*HPr (**left panels**) and *Bs*HPr (**right panels**) proteins at 302, 333, 362, 400, and 450 K. One protein conformation is given for each secondary structure profile to visualize the structural unfolding of both proteins. The color codes are the same as in Figure 1.

**Figure 3 ijms-24-09557-f003:**
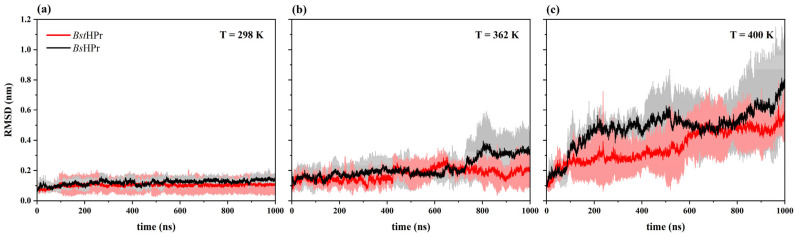
RMSD trajectory analyses of the *Bst*HPr (red line) and *Bs*HPr (black line) proteins at: (**a**) 298 K, (**b**) 362 K, and (**c**) 400 K. The solid lines represent the average RMSD value and the shaded regions denote the respective SD from three independent simulations. The three independent trajectories and the average values for the set of temperatures are available in Appendix A.

**Figure 4 ijms-24-09557-f004:**
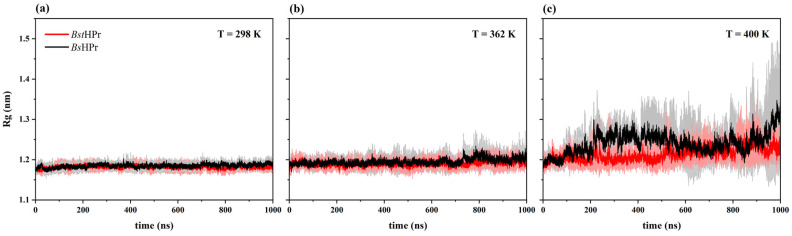
Rg trajectory analyses of the *Bst*HPr (red line) and *Bs*HPr (black line) proteins at: (**a**) 298 K, (**b**) 362 K, and (**c**) 400 K. The solid lines represent the average Rg values and the shaded regions denote the respective SD from three independent simulations. The three independent trajectories and the average values for the set of temperatures are available in Appendix A.

**Figure 5 ijms-24-09557-f005:**
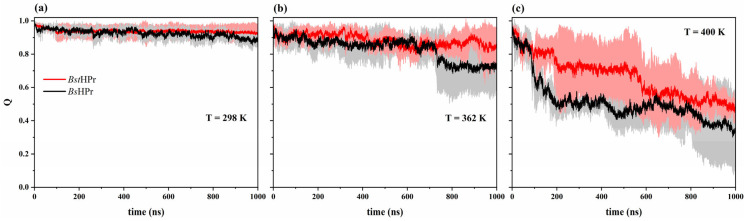
Time evolution of the fraction of native contacts for the *Bst*HPr (red line) and *Bs*HPr (black line) proteins at: (**a**) 298 K, (**b**) 362 K, and (**c**) 400 K. The solid lines represent the average Q value and the shaded areas denote the respective SD from three independent simulations. The three independent trajectories and the average values for the set of temperatures are available in Appendix A.

**Figure 6 ijms-24-09557-f006:**
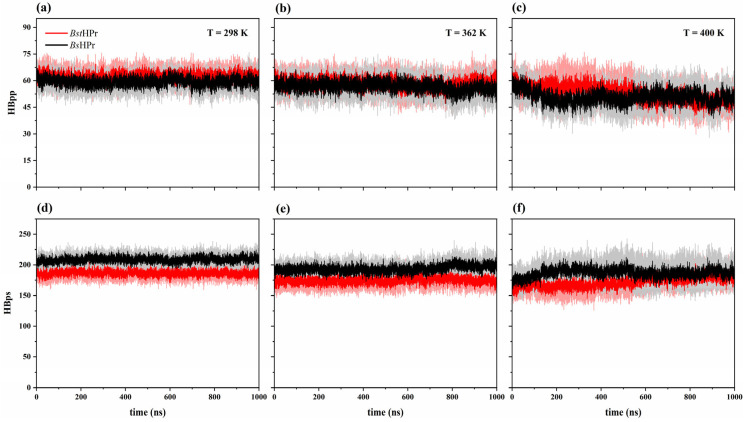
Time evolution of the hydrogen bonds for the *Bst*HPr (red line) and *Bs*HPr (black line) proteins at: (**a**) 298 K, (**b**) 362 K, and (**c**) 400 K for HBpp, and (**d**) 298 K, (**e**) 362 K, and (**f**) 400 K for HBps. The solid lines represent the average HB values and the shaded regions denote the respective SD from three independent simulations. The three independent trajectories and the average values for the set of temperatures are available in Appendix A.

**Figure 7 ijms-24-09557-f007:**
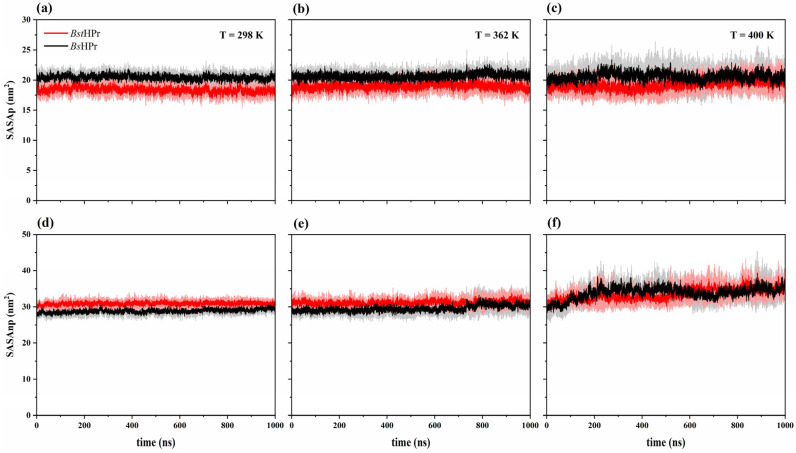
Time evolution of the solvent accessible surface area for the *Bst*HPr (red line) and *Bs*HPr (black line) proteins at: (**a**) 298, (**b**) 362, and (**c**) 400 K for SASAp, and (**d**) 298, (**e**) 362, and (**f**) 400 K for SASAnp. The solid lines indicate the average SASA values and the shaded regions denote the respective SD from three independent simulations. The three independent trajectories and the average values for the set of temperatures are available in Appendix A.

**Figure 8 ijms-24-09557-f008:**
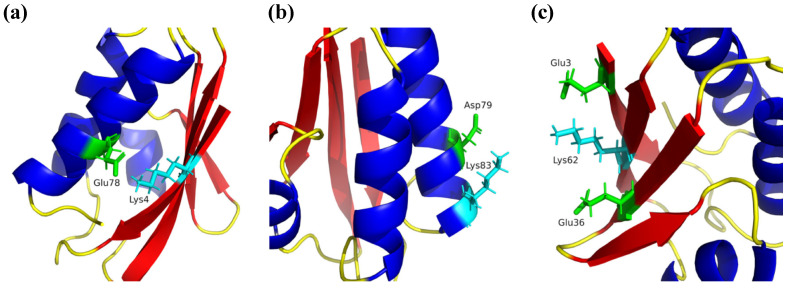
Cartoon diagrams showing residue pairs forming salt bridges in both proteins: (**a**) pairwise Glu78-Lys4 of the *Bs*HPr protein, (**b**) pairwise Glu78-Lys4 of the *Bst*HPr protein, and (**c**) triad formed by Glu3–Lys62–Glu36 in the *Bst*HPr protein. Structures were built at t = 0 ns of simulation at 298 K. The red, blue, and yellow colors indicate the β-strand, α-helix, and random coil structures, respectively. Acid and basic residues are colored in green and cyan colors, respectively.

**Figure 9 ijms-24-09557-f009:**
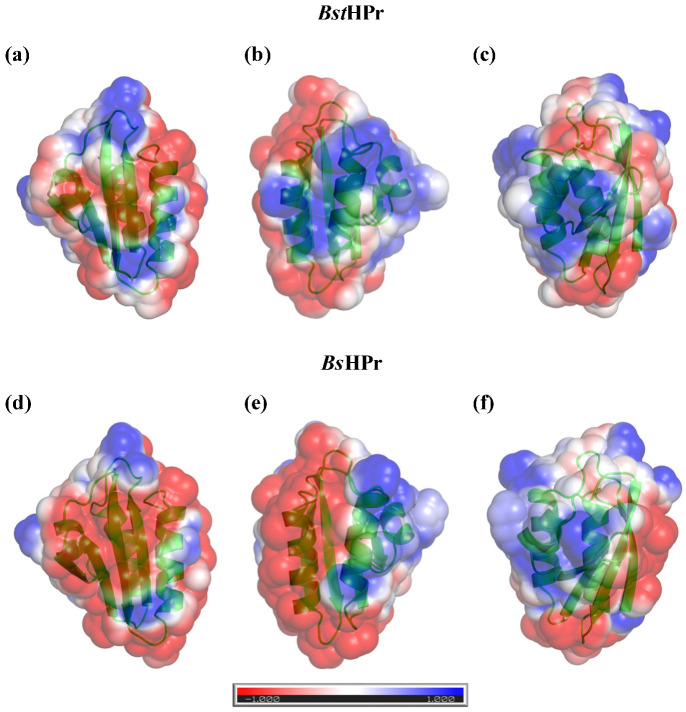
Electrostatic potential maps of the surface of *Bst*HPr (**top** panels) and *Bs*HPr (**bottom** panels) proteins. The distribution of charge is shown on three different structural surfaces: (**a**,**d**) indicate the β-sheet; (**b**,**e**) correspond to α-helices; and (**c**,**f**) denote the interphase β-sheet/α-helices. Positive and negative charges are depicted in blue and red colors, respectively. The charge gradient is shown in the lower bar. Structures were made at t = 0 ns of simulation at 298 K.

**Figure 10 ijms-24-09557-f010:**
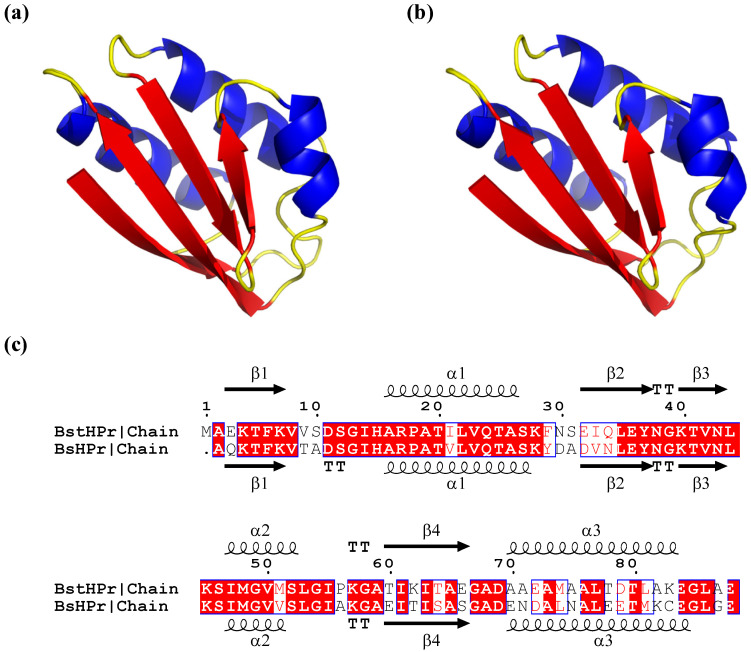
Cartoon representation of tertiary structure for (**a**) *Bst*HPr and (**b**) *Bs*HPr proteins at *t* = 0 ns of simulation at 298 K. The sequence alignment based on the primary structure of both proteins is shown in (**c**): identical amino acids are evidenced in red color and similar amino acids are indicated with blue boxes and red background. The red, blue, and yellow colors indicate the β-strand, α-helix, and random coil structures in (**a**,**b**), respectively. Structures were processed with PyMOL version 2.5.2 program (https://pymol.org (accessed 21 April 2023)). The sequence alignment was made up with ESPript 3.0 program (http://espript.ibcp.fr (accessed 10 March 2023)) [39].

**Table 1 ijms-24-09557-t001:** RMSD average and SD values for *Bst*HPr and *Bs*HPr proteins.

T (K)	*Bst*HPr		*Bs*HPr	
	RMSD (nm)			
	Avg	SD	Avg	SD
298	0.102	±0.040	0.121	±0.024
333	0.163	±0.048	0.167	±0.064
362	0.173	±0.058	0.214	±0.095
400	0.361	±0.146	0.483	±0.179
450	1.133	±0.437	1.225	±0.322

**Table 2 ijms-24-09557-t002:** Rg average and SD values for *Bst*HPr and *Bs*HPr proteins.

T (K)	*Bst*HPr		*Bs*HPr	
	Rg (nm)			
	Avg	SD	Avg	SD
298	1.181	±0.007	1.185	±0.007
333	1.195	±0.013	1.193	±0.013
362	1.190	±0.010	1.195	±0.014
400	1.213	±0.023	1.241	±0.046
450	1.389	±0.228	1.415	±0.188

**Table 3 ijms-24-09557-t003:** Native contacts average and SD values for *Bst*HPr and *Bs*HPr proteins.

T (K)	*Bst*HPr		*Bs*HPr	
	Q			
	Avg	SD	Avg	SD
298	0.938	±0.029	0.925	±0.029
333	0.887	±0.047	0.870	±0.073
362	0.884	±0.054	0.832	±0.101
400	0.662	±0.164	0.513	±0.162
450	0.199	±0.173	0.142	±0.131

**Table 4 ijms-24-09557-t004:** HBpp and HBps average and SD values for *Bst*HPr and *Bs*HPr proteins.

T (K)	*Bst*HPr		*Bs*HPr		*Bst*HPr		*Bs*HPr	
	HBpp				HBps			
	Avg	SD	Avg	SD	Avg	SD	Avg	SD
298	62.4	±3.7	59.4	±3.8	186.7	±7.2	208.6	±7.6
333	60.2	±4.2	57.2	±4.4	181.7	±8.2	201.9	±8.9
362	58.5	±4.6	56.8	±4.6	174.8	±8.7	193.5	±9.5
400	53.1	±6.5	50.7	±5.7	172.0	±12.5	186.9	±15.0
450	41.6	±7.3	41.7	±6.7	174.4	±14.5	187.3	±13.9

**Table 5 ijms-24-09557-t005:** SASAp and SASAnp average and SD values for *Bst*HPr and *Bs*HPr proteins.

T (K)	*Bst*HPr		*Bs*HPr		*Bst*HPr		*Bs*HPr	
	SASAp (nm^2^)				SASAnp (nm^2^)			
	Avg	SD	Avg	SD	Avg	SD	Avg	SD
298	18.37	±0.72	20.43	±0.68	30.89	±0.74	28.84	±0.80
333	18.88	±0.80	20.66	±0.78	31.50	±0.99	29.41	±1.29
362	18.97	±0.82	20.64	±0.81	31.17	±1.04	29.51	±1.35
400	19.24	±1.14	20.58	±1.45	33.47	±2.08	33.90	±2.38
450	20.89	±1.94	22.06	±1.98	42.93	±5.94	43.22	±5.13

**Table 6 ijms-24-09557-t006:** Average frequencies of salt bridges of *Bs*HPr protein at different temperatures.

Residue Pairs	Temperature (K)				
	298	333	362	400	450
Glu78–Lys4	0.906	0.849	0.906	0.408	0.106
Glu84–Arg17	0.570	0.489	0.570	0.412	0.056
Asp11–Lys57	0.516	0.404	0.516	0.136	0.032

**Table 7 ijms-24-09557-t007:** Average frequencies of salt bridges of *Bst*HPr protein at different temperatures.

Residue Pairs	Temperature (K)				
	298	333	362	400	450
Asp79–Lys83	0.822	0.773	0.773	0.527	0.341
Glu84–Arg17	0.579	0.495	0.526	0.292	0.104
Asp11–Lys57	0.412	0.397	0.396	0.220	0.081
Glu3–Lys62	0.620	0.617	0.633	0.597	0.088
Glu36–Lys62	0.380	0.403	0.526	0.553	0.203

**Table 8 ijms-24-09557-t008:** Simulation conditions for *Bs*HPr and *Bst*HPr proteins.

Condition	*Bst*HPr	*Bs*HPr
Cell volume (nm^3^)	164.59	145.84
Protein (atoms)	1290	1268
Water (atoms)	15,084	13,131
Ions (Cl^−^)	1	1
Ions (Na^+^)	2	5

## Data Availability

Data are available from the corresponding author on reasonable request.

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
