# Peer review of "Molecular Dynamics Simulations of HPr Proteins from a Thermophilic and a Mesophilic Organism: A Comparative Thermal Study"

_ijms, 2023, doi:10.3390/ijms24119557_

Round 1
Reviewer 1 Report
Summary:
The manuscript submitted by Gómez-Flores et al. presents a comparative analysis of mesophilic and thermophilic histidine-containing phosphocarrier (HPr). Using 1 us molecular dynamic simulation data, authors present temperature-dependent changes in protein secondary structure, intermolecular interaction, salt bridges, SASA (polar and non-polar), and others. The paper is recommended for publication after the author addresses the queries listed below and has made the required changes in the manuscript.
Comments
1) Given the melting temperature of mesophilic protein (74 C) being lower than that of thermophile (89 C). I would like the authors to explain the loss of the α-2 structure of BstHPr at 333 K (60 C) much below its melting temperature.
2) For the simulation at 362 K (90 C), the BsHPr shows that most of the protein secondary structure doesn’t change, despite the simulation temperature being 15 C higher than the melting temperature.
3) Why does the radius of gyration and native contact for BsHPr not change significantly at 362 K (90 C) despite being 15 C higher than the melting temperature?
4) For BsHPr, the secondary structure shows a major change at 400 K, however, SASAnp does not reflect a major change going from 362K to 400 K (Table 6). I would suggest the author show this SASAnp in the 3D structure of both proteins.
5) Moreover, I would also recommend analysing the presence of ILV clusters (https://proteintools.uni-bayreuth.de/) in both the protein and how this cluster changes with temperature.
Reviewer 2 Report
The work focuses on the thermal stability of the thermophilic protein BstHPr using molecular dynamics simulations at five different temperatures. Their work found mesophilic protein is the most affected one. The conclusion is well-supported by the analysis. I recommend this work to be accepted in its present form.
Author Response
We would like to thank the reviewer.